# Antivirals Against Chikungunya Virus: Is the Solution in Nature?

**DOI:** 10.3390/v12030272

**Published:** 2020-02-29

**Authors:** Daniel Oliveira Silva Martins, Igor de Andrade Santos, Débora Moraes de Oliveira, Victória Riquena Grosche, Ana Carolina Gomes Jardim

**Affiliations:** 1Laboratory of Virology, Institute of Biomedical Science, ICBIM, Federal University of Uberlândia, Uberlândia, MG 38408-100, Brazil; danielosmartins@gmail.com (D.O.S.M.); igoras244@gmail.com (I.d.A.S.); deboramoraaes@hotmail.com (D.M.d.O.); victoriagrosche@live.com (V.R.G.); 2São Paulo State University, Institute of Biosciences, Letters and Exact Sciences (IBILCE), State University of São Paulo, São José do Rio Preto, SP 15054-000, Brazil

**Keywords:** chikungunya virus, antiviral, natural compounds

## Abstract

The worldwide outbreaks of the chikungunya virus (CHIKV) in the last years demonstrated the need for studies to screen antivirals against CHIKV. The virus was first isolated in Tanzania in 1952 and was responsible for outbreaks in Africa and Southwest Asia in subsequent years. Between 2007 and 2014, some cases were documented in Europe and America. The infection is associated with low rates of death; however, it can progress to a chronic disease characterized by severe arthralgias in infected patients. This infection is also associated with Guillain–Barré syndrome. There is no specific antivirus against CHIKV. Treatment of infected patients is palliative and based on analgesics and non-steroidal anti-inflammatory drugs to reduce arthralgias. Several natural molecules have been described as antiviruses against viruses such as dengue, yellow fever, hepatitis C, and influenza. This review aims to summarize the natural compounds that have demonstrated antiviral activity against chikungunya virus in vitro.

## 1. Introduction

Chikungunya fever is a tropical disease caused by the chikungunya virus (CHIKV) which is transmitted to humans by the bite of an infected mosquito of *Aedes* sp. The first case of chikungunya fever was reported in 1952 in Tanzania [1]. In February 2005, a major outbreak of chikungunya occurred on the islands of the Indian Ocean [2]. A large number of cases occurred in Europe and India in 2006 and 2007, respectively [2]. Several other countries in Southeast Asia were also affected [3]. In December 2013, autochthonous cases were confirmed in the French part of the Caribbean island of St Maarten [4]. Since then, local transmission has been confirmed in over 60 countries in Asia, Africa, Europe, and the Americas. In 2014, more than 1 million suspected cases were reported in the Americas, with 1,379,788 suspected cases and 191 deaths in the Caribbean islands, Latin American countries, and the United States of America (USA) [5]. Canada, Mexico, and USA have also recorded imported cases. The countries reporting the most cases were Brazil (265,000 suspected cases), and Bolivia and Colombia (19,000 suspected cases each) [6]. The first autochthonous transmission of chikungunya reported in Argentina occurred in 2016 following an outbreak of more than 1000 suspected cases [7]. In the African region, Kenya reported an outbreak of chikungunya resulting in more than 1700 suspected cases. In 2017, Pakistan continues to respond to an outbreak which started in 2016 [8]. These virus outbreaks have raised concerns on studies of CHIKV epidemiology and antiviral research [9].

CHIKV belongs to the Alphavirus genus and the *Togaviridae* family. It is a positive-sense, single-stranded RNA (12 kb in length) virus, with an enveloped icosahedral capsid [10]. The virus lifecycle starts via the attachment of the viral glycoproteins to the cell membrane receptors, mainly to MXRA8 [11,12] but also to prohibitin (PHB) [13], phosphatidylserine (PtdSer) [14], and glycosaminoglycans (GAGs) [15] receptors in mammalian and to ATP synthase β in mosquito cells [16], forming a pore. Then, a virus capsid is released into the cytoplasm, where the replication process takes place. Viral genome is uncoated and directly translated into nonstructural (NS) proteins nP1–4. The NS proteins form the viral replicase complex that catalyzes the synthesis of a negative strand, a template to synthesize the full-length positive sense genome, and the subgenomic mRNA. The subgenomic mRNA is translated in a polyprotein, which is cleaved to produce the structural proteins C, E3, E2, 6k, and E1, followed by the assembly of the viral components and virus release (Figure 1) [17,18].

Chikungunya fever is characterized by strong fever, arthralgia, backache, headache, and fatigue. In some cases, cutaneous manifestation and neurological complications can occur [19,20]. There is no Food and Drug Administration (FDA) approved specific antiviral or vaccine against CHIKV. Therefore, the treatment of infected patients is based on palliative care, using analgesics for pain and non-steroidal anti-inflammatory drugs to reduce arthralgia in chronic infections [10].

Due to the lack of efficient anti-CHIKV therapy, researches have been developed to identify new drug candidates for the future treatment of chikungunya fever [21]. Among them, antiviral research based on natural molecules is a potential approach. Many natural compounds showed antiviral activity against a variety of human viruses such as dengue (DENV) [22,23,24,25], yellow fever (YFV) [25,26,27], hepatitis C (HCV) [28,29,30,31,32], influenza [33,34], and zika (ZIKV) [33,35,36]. Here, we aim to summarize the natural compounds previously described to possess anti-CHIKV activity (Table 1).

## 2. Inhibitors of CHIKV Replicative Cycle

### 2.1. Epigallocatechin Gallate (Green Tea)

Epigallocatechin gallate (EGCG) is the major catechin constituent in green tea that has shown antiviral activity against CHIKV in vitro [37]. HEK 293T cells (human kidney cells) were infected with the pseudo particles CHIKV-mCherry-490 with a multiplicity of infection of 1 (MOI = 1) in the presence or absence of EGCG at 10 μg/mL, which blocked up to 60% of CHIKV entry. Through lentiviral expression of CHIKV glycoprotein, the authors evaluated the antiviral activity of EGCG on entry steps and suggested that EGCG interferes with CHIKV entry due to their effect on CHIKV envelope protein [37].

### 2.2. Chloroquine

According to the studies of Khan and coworkers, a synthetic compound derived from the natural Chloroquine used to treat malaria infection has shown antiviral activity against CHIKV [38]. To do this, Vero cells were infected with the African East-Central-South (ECSA) CHIKV genotype, DRE-06 strain, and incubated with the compound at 5, 10, or 20 μM to evaluate its antiviral activity. Three treatment strategies were used for the plaque assay: (1) pretreatment of the cells 24 h before infection; (2) concurrent treatment by simultaneously adding virus and chloroquine; and (3) treatment of cells up to 6 h post-CHIKV infection of Vero cells. Chloroquine at 20 μM was nontoxic to the cells and inhibited CHIKV entry by approximately 94% when cells were pretreated, 70% in the concurrent treatment, and 65% in the post-infection treatment. The results suggested that this compound presents strong antiviral activity, mainly when administered 24 h prior to infection [38].

### 2.3. Apigenin, Chrysin, Luteonin, Narigerin, Silybin, and Prothipendyl

Pohjala and colleagues demonstrated the anti-CHIKV activity of five natural compounds by using either a replicon cell line expressing the nonstructural proteins of CHIKV and the *eGFP* and Renilla luciferase (*Rluc)* markers or the full-length virus genetically modified with the reporter *Rluc*. Firstly, BHK21 (baby hamster kidney) cells were infected with the full length CHIKV-*Rluc* (MOI = 0.001) and simultaneously treated with different concentrations of each compound ranging from 0.01 to 100 µM for 16 h. The compounds apigenin (inhibitory concentration (IC_50_) = 70.8 µM), chrysin (IC_50_ = 126.6 µM), narigenin (IC_50_ = 118.4 µM), silybin (IC_50_ = 92.3 µM), and prothipendyl (IC_50_ = 97.3 µM) significantly inhibited CHIKV-*Rluc* replication [39].

In addition, Muralli and coworkers also tested the antiviral activity of apigenin and luteonin ethanolic fraction from *Cynodon dactylon* in Vero cells and found that the fractions inhibited 98% of CHIKV activity at concentration of 50 µg/mL through the cytopathic effect [40]. Using a reverse transcriptase polymerase chain-reaction (RT-PCR) the authors also demonstrated that virus RNA levels decreased under treatment. In another study, apigenin and luteonin were isolated from a fraction of the *Cynodon dactylon* plant, obtained from the National Institute of Virology of India, and were used to assess the cytotoxicity and antiviral activity in Vero cells. Results showed that concentrations ranging from 5 to 200 μg/mL were nontoxic as determined by 3-(4,5-dimethylthiazol-2-yl)-2,5-diphenyltetrazolium bromide cell proliferation assay (MTT assay). In addition, treatment of cells at 10, 25, and 50 μg/mL showed a reduction of viral activity by decreasing 68%, 88%, and 98% of the cytopathic effect of the virus, respectively [39,40].

### 2.4. Flavaglines

As CHIKV uses prohibitin as a receptor to entry into mammalian cells [13], Wintachai and colleagues investigated the anti-CHIKV activity of the plant-derived compounds sulfonyl amidines 1M and the flavaglines FL3 and FL23 [41], previously reported to interact with this receptor. These compounds demonstrated antiviral activity against the CHIKV strain E1:226V East-Central-South-Africa (ECSA) genotype of a Thai isolate. The cell line HEK-293T/17 was added to each compound at specific concentrations (1, 5, 10, and 20 nM) for one hour and then infected with 10 pfu/cell of CHIKV. After 20 h, cell pellets were submitted to flow cytometry and the supernatant to a plaque assay to measure CHIKV titers. All three compounds significantly reduced the percentage of viral production in the infected cells at 10 and 20 nM concentrations. Sulfonyl amidine 1M and FL23 at 20 nM reduced viral cytopathic effect by approximately 40%, and FL3 at 20 nM reduced viral yield by 50% [41].

### 2.5. Compounds from Tectona grandis

The antiviral activity of three isolated and characterized compounds from *Tectona grandis* had its antiviral activity tested against the CHIKV strains ECSA KC 969208 and Asian KC969207 in Vero cells [42]. The authors determined IC_50_ of the compounds 2-(butoxycarbonyl) benzoic acid (BCB), 3,7,11,15-tetramethyl-1-hexadecanol (THD), and benzene-1-carboxylic acid-2-hexadeconate (BHCD). They demonstrated that the most potent anti-CHIKV activity was observed for BHCD with selectivity index (SI) of 116 for the Asian strain and 4.66 for ECSA. In silico analyses were performed and showed that the compound possessed strong interactions with CHIKV envelope protein 1 (E1) and poor interactions with nonstructural proteins (nSP) that may suggest that this compound could act on CHIKV entry [42].

### 2.6. Trigocherrierin A

The work of Bourjot and colleagues showed that compounds isolated from the *Trigonostemon cherrieri* presented inhibitory activity against CHIKV replication [43]. Vero cells were used in cell proliferation assay (MTS) to evaluate the anti-CHIKV activity of compounds by decreasing the cell death induced by the virus infection [43]. Among the isolated compounds, trigocherrierin A inhibited death of cells caused by the virus with a concentration that induced half of the maximum effect (EC_50_) of 0.6 ± 0.1 μM, CC_50_ of 43 ± 16 μM, and the SI of 71.7. Thus, trigocherrierin A has been shown to be the most potent tested compound against CHIKV replication in this study [43].

### 2.7. Harringtonine

Harringtonine, a natural compound derived from the Japanese plant *Cephalotaxus harringtonia*, demonstrated antiviral activity against CHIKV replication [44]. The authors investigated the anti-CHIKV activity of this compound by using the cell lines BHK-21, C6/36 (embryonic tissue cells of the *Aedes albopictus* mosquito), and HSMM (human skeletal muscle myoblasts) and the virus strains CHIKV-0708 (Singapore 07/2008, lacking the A226V mutation in E1 protein) and CHIKV-122508 (SGEHICHD 122508, having the A226V mutation in the E1 protein) [44]. In BHK-21 cells, harringtonine at 1 and 10 μM showed potent anti-CHIKV action, inhibiting up to 90% of viral replication with cell viability higher than 80%. Aiming to investigate the harringtonine mechanism of action, the authors performed a time addition assay. Compounds were added at different concentrations, prior to infection (−2 h) and at 0, 2, 6, 12, and 16 h post infection (h.p.i.). Treatments showed inhibition of CHIKV replication at 2 h.p.i, indicating that harringtonine inhibits the early steps of the CHIKV replicative cycle. Additionally, cells were infected and treated for 6 h, and western blot and qRT-PCR assays were performed. The results showed that harringtonine reduced negative- and positive-sense RNAs of CHIKV and the production of nSP3 and E2 proteins [44].

### 2.8. Diterpene Ester (phorbol-12,13-didecanoate)

Twenty-nine diterpenoids isolated from *Euphorbiaceae* species had their antiviral activity tested against CHIKV (Indian Ocean strain 899) in vitro through MTS assay [45,46]. First, media with serial dilutions of each compound was added to empty 96-well microplate, and then, each well was added of media containing Vero cells (2.5 × 10^3^ cells per well) and CHIKV for 6–7 days. Among the tested compounds, phorbol-12,13-didecanoate was shown to be the strongest candidate as an antivirus against CHIKV replication, with an EC_50_ 6.0 ± 0.9 nM [45,46].

### 2.9. Daphanane Diterpenoid Ortho Esters

A panel of diterpenoids or thioesters isolated from *Trigonostemon cherrieri* was used to evaluate the antiviral activity against CHIKV [47]. Vero cells were used to determine the cytotoxicity of compounds, and antiviral properties were accessed by plaque assay. Among the tested compounds, Trigoocherrins A, B, and F were shown to be potent inhibitors of CHIKV replication with SIs of 23, 36, and 8, respectively [45].

### 2.10. Aplysiatoxin-Related Compounds

Five bioactive compounds from the cyanobacteria *Trichodesmiumery thraeuma* had their antiviral activity evaluated [47]. Cell viability was measured and a dose-dependent anti-CHIKV assay was performed to access the antiviral activity of the compounds under pre- or post-treatment conditions. The Debromo analogues 2 and 5 showed significant antiviral activity in post-treatment of infected BHK 21 cells with EC_50_ of 1.3 and 2.7 μM and SI of 10.9 and 9.2, respectively. The authors suggested that the antiviral activity of these compounds blocks the replication step of the CHIKV replicative cycle [47].

### 2.11. Tannic Acid

Tannic acid (TA) is a compound found in different species of plants, but its structure varies according to their sources. It previously demonstrated antiviral activity against viruses as Herpes (HSV) and HCV [48,49]. The anti-CHIKV activity of TA was investigated by KONISHI and HOTTA by performing plaque reduction assay using BHK-21 cells [50]. TA reduced 50% of the virus infectivity in lower concentrations and demonstrated inhibition of virus post-entry steps in BHK-21 cells. To investigate which chemical group of TA is associated with its antiviral activity, the authors tested TA analogues on their virus-inhibiting capacities. The results demonstrated that phenolic hydroxyl groups may be related to the antiviral activity, since the displacement of these groups make the molecule ineffective [50].

### 2.12. Silymarin

Silymarin is a polyphenolic compound from flavonoids family, is extracted from *Silybum marianum*, and is described to possesses antiviral activity against HCV [51]. A study tested the activity of silymarin on CHIKV genotype ECSA with A226V mutation in E1 protein from a clinical strain isolated in an outbreak in 2008. BHK-21 and Vero cells were used to evaluate different steps of the viral replicative cycle, and silymarin showed inhibition of post-entry stages of CHIKV with an EC_50_ of 16.9 µg/mL and SI of 25.1. By using a stable cell line expressing CHIKV replicon and *EGFP* and *Rluc* markers [39], it was demonstrated that silymarin suppressed 93.4% of CHIKV replication. Western blot assay was performed, showing that silymarin treatment decreased the amounts of nSP1, nSP3, and E2 proteins [52].

### 2.13. Baicalein, Fisetin, and Quercetagetin

Baicalein, fisetin, and quercetagetin are compounds from the flavonoids family that exhibited antiviral activity against DENV [22] and enterovirus A71 [53]. Lani and colleagues infected Vero cells with the CHIKV genotype ECSA strain from the outbreak of 2008 and evaluated their effects in reducing the cytopathic effect resulting from viral infection [54]. All three compounds were found to inhibit CHIKV replication in a dose-dependent manner and reduced E2, nSP1, and nSP3 protein synthesis, as showed by Western blot analysis. Baicalein and quercetagetin showed anti-CHIKV activity by inactivating the virus, preventing the attachment of the virus to the host cells and blocking post-entry stages, with EC_50_ of 1.891 µg/mL and 13.85 µg/mL, respectively. Fisetin only inhibited post-entry steps with EC_50_ of 8.44 µg/mL [54].

### 2.14. Bryostatin

Bryostatin is a macrolide lactone derived from a marine animal named Bugula neritina [55]. It was described by the antineoplastic activity [56], affects Alzheimer’s disease [57], and has been related to the eradication of human immunodeficiency virus reservoirs [58]. The anti-CHIKV activities of the Bryostatin analogs salicylate-derived analog 1, C26-capped analog 2, and C26-capped analog 3 were assessed by evaluating the cytopathic effect (CPE) caused by CHIKV Indian Ocean lineage strain 899 replication under treatment with these three compounds [59]. All of the Bryostatin analogs inhibited the CHIKV replicative cycle, decreasing infectious progeny and viral RNA copies, confirmed by supernatant titration and RT-PCR. A time-addition assay showed that these compounds inhibited late stages of CHIKV replication, with EC_50_ rates of 4 µM, 8 µM, and 7.5 µM, respectively. Additionally, salicylate-derived analog 1 but not the other compounds blocked entry of CHIKV pseudoparticles into Buffalo green monkey kidney cells (BGM) [59].

### 2.15. Prostatin

Bourjot and coworkers described the effect of prostratin, a compound derived from *Trigonostemon howii*, on CHIKV infection in Vero cells by a CPE assay (EC_50_ = 2.6 µM) [60]. Another work used CHIKV lineage Indian Ocean 899 to infected Vero, BGM, or Human embryonic lung fibroblasts (HEL) cells at MOI of 0.001 under the treatment with prostratin and obtained EC_50_ of 8 µM, 7.6 µM, and 7.1 µM, respectively. Using a delay treatment associated with a RT-PCR or CHIKV pseudoparticle techniques, it was demonstrated that prostratin decreased both the number of CHIKV genome copies and the production of infectious progeny virus particles. A western blot assay was used to detect CHIKV proteins and showed that prostratin also reduced the accumulation of nSP1 and capsid proteins [60].

### 2.16. Berberine

Berberine is a compound found in plants from the *Berberis* genus, family *Berberidaceae*, that previously demonstrated antiviral activity against other viruses [61]. Varghese and colleagues analyzed the antiviral effect of berberine on the CHIKV replication cycle using the CHIKV lineage LR2006 OPY1 with the *Rluc* marker to infect HEK-293T, HOS (humam bone osteosarcoma), and CRL-2522 cells. The berberine EC_50_ for each cell line were 4.5, 12.2, and 35.3 µM, respectively. This compound was also active against the different CHIKV strains LR2006 OPY1, SGP11, and CNR20235, showing EC_50_ of 37.6, 44.2, and 50.9 µM, respectively. Berberine showed no inhibition on CHIKV entry or replication but decreased viral RNA and viral protein synthesis, suggesting that berberine is indirectly perturbing CHIKV replication by affecting host components [61].

### 2.17. Avermectin Derivates

Avermectin is naturally produced in *Streptomyces avermitilis* bacteria and showed different biological properties including antiparasitic [62], antiviral [63], and antibacterial [64,65] activities. Ivermectin (IVN) and abamectin (ABN) are chemically modified derivatives of avermectin. The activity of these derivatives on the CHIKV replication cycle was described in a study that used BHK-21 with CHIKV containing the *Rluc* gene [66]. IVN and ABN demonstrated EC_50_ of 0.6 µM and 1.5 µM, respectively, and strongly reduced nSP1 and nSP3 even in high MOIs. A time-of-addition assay demonstrated that IVN and ABN interfered in earlier stages of CHIKV cycle but not when cells were pretreated. Alternatively, the activity of these compounds was decreased in the later stages of the CHIKV replicative cycle [66].

## 3. Prospects

The aim of this review was to summarize data from literature concerning the natural compounds described to possess anti-CHIKV activity. Altogether, data is heterogeneous since authors developed a variety of assays using different cell lines and CHIKV strains or replicons. Some studies did not elucidate the mechanism of action (MOA) of the compound, retaining their information as EC_50_, CC_50_, and/or SI. For most of the compounds presented in this review, it would be desirable to demonstrate the MOA in order to elucidate the biochemical and molecular basis of the compound–virus or compound–cell interactions and to be able to predict and promote strategies for pharmacological outcomes in further studies [67]. Also, the investigation of the effects of each compound in different cell lines would provide important information concerning the effects of these compounds on the host cells [68,69]. Besides that, all data summarized here represent a relevant source of knowledge concerning the antiviral potential of molecules isolated from nature.

From the natural compounds cited in this review, chloroquine was the only compound tested in vivo, in non-human primates, and in human clinical trials. Chloroquine is already used for the treatment of malaria [70]. However, despite the in vitro results, chloroquine demonstrated no relevant results in vivo in decreasing viremia or in reducing clinical manifestations during acute stage of CHIKV infection [71]. Therefore, the results demonstrated by in vitro analysis were not correlated with the in vivo analysis that showed that chloroquine was not suitable for patients with CHIKV. Additionally, the remaining compounds described here have not been tested in vivo yet, representing a delay in anti-CHIKV drug development.

Apart from the chloroquine case, all compounds that demonstrated antiviral activity have the potential to be further investigated by their therapeutically properties against chikungunya fever. Furthermore, natural compounds may present as a source of molecules with potent biological activities that could be used as templates to the development of novel antivirals.

## 4. Conclusions

The spread of CHIKV in the last years demonstrated the need to develop effective antiviruses to treat chikungunya fever and to prevent future outbreaks. In this context, natural compounds have shown potent antiviral activity against a range of viruses. This review summarized the natural compounds described to possess anti-CHIKV activity by blocking early and/or late stages of virus replication in vitro. Apart from the great antiviral activity of the described compounds, further research is needed for the development of future treatments.

## Figures and Tables

**Figure 1 viruses-12-00272-f001:**
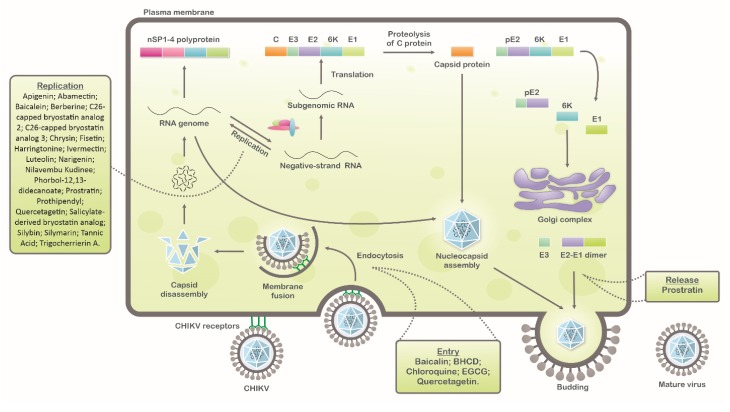
Schematic representation of chikungunya virus (CHIKV) replication cycle: Natural compounds with antiviral activity against CHIKV are indicated in each step of virus replication cycle (entry, replication, and release).

**Table 1 viruses-12-00272-t001:** Natural compounds with antiviral activity against CHIKV.

Compound	Structure	Inhibition	SI or EC_50_	Cell Line
Abamectin [66]	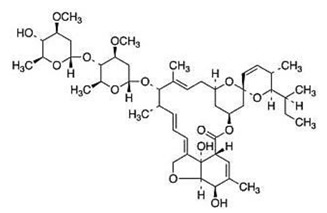	Replication	1.5 µM	BHK-21
Apigenin [39,40]	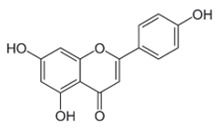	Infection/Replication	70.8 µM	BHK 21
Baicalein [54]	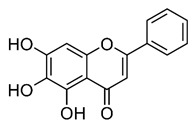	Infection and replication	1.891 µg/mL	BHK-21
Baicalein [54]	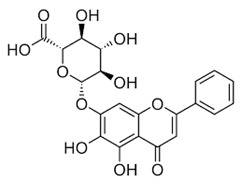	Entry, binding	6.997 µM	Vero
Berberine [61]	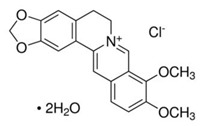	Replication (interfering in host components)	≤35.3 µM	CRL-2522, HEK-293T, and HOS
BHCD [42]	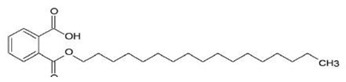	Entry	116 (Asian strain) and 4.66 (ECSA)	Vero and in silico
C26-capped bryostatin analog 2 [59]	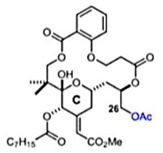	Replication	8 µM	Vero
C26-capped bryostatin analog 3 [59]	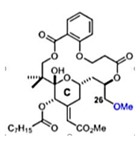	Replication	7.5 µM	Vero
Chloroquine [38]	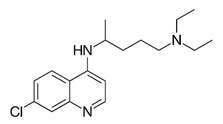	Entry	37.14	Vero
Chrysin [39]	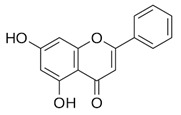	Infection	126.6 µM	BHK 21
EGCG [37]	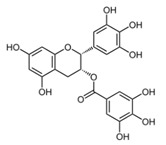	Entry steps; cell attachment	6.54 µg/mL	HEK 293T
Fisetin [54]	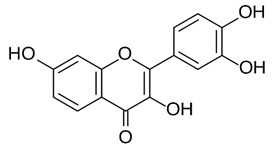	Replication	8.44 µg/mL	BHK-21
Harringtonine [44]	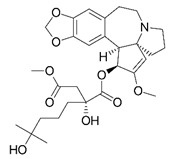	Early stages of replication	0.24 µM	BHK 21
Ivermectin [66]	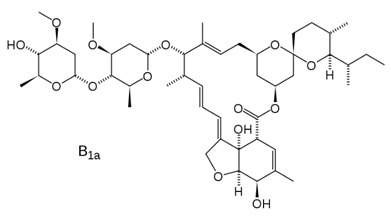	Replication	0.6 µM	BHK-21
Luteolin [40]	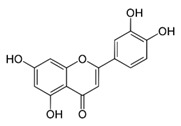	Replication	NS	Vero
Narigenin [39]	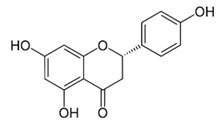	Infection	118.4 µM	BHK 21
Prostratin [60]	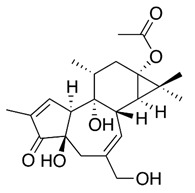	Replication and release	2,6 µM and ± 8 µM	Vero, BGM, and HEL
Prothipendyl [39]	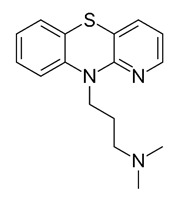	Replication	97.3 µM	BHK 21
Quercetagetin [54]	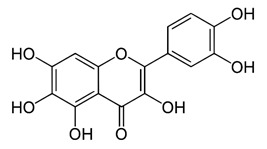	Entry and binding	43.52 µM	Vero
Quercetagetin [54]	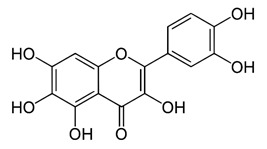	Entry and replication	13.85 µg/mL	BHK-21
Salicylate-derived bryostatin analog [59]	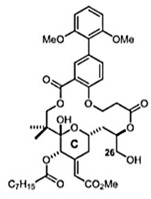	Entry and replication	4 µM	Vero
Silybin [39]	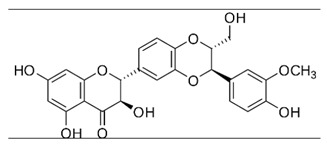	Infection	92.3 µM	BHK 21
Silymarin [52]	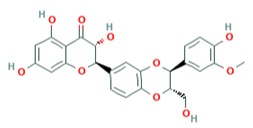	Replication	16.9 µg/mL	BHK-21 and Vero
Tannic Acid [50]	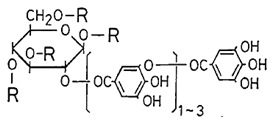	Replication	NS	BHK-21
Phorbol-12,13-dideca-noate [46]	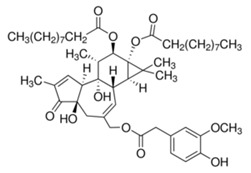	Replication	6 ± 0.9 nM	Vero
Trigocherrierin [43]	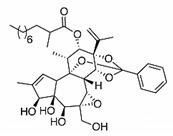	Replication	0.6 ± 0.1 µM	Vero

NS = Not shown, data not shown.

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
