# Peer review of "Antivirals Against Chikungunya Virus: Is the Solution in Nature?"

_viruses, 2020, doi:10.3390/v12030272_

Round 1

Reviewer 1 Report

This manuscript reads more like a summary of specific studies than a compendium of work that has been done to investigate antiviral properties of natural compounds. Rather than summarising data (shown in the manuscript via extensive duplication of experimental parameters that may not be relevant to the issue overall, ie. description of MOIs, use of GFP reporter constructs), the authors should instead focus on the important ones, and if even tentatively describe the steps required, or the work being done to translate this work in vivo.

Natural compounds have long been hailed as promising drug candidates - and they may well be - however at no point do the authors acknowledge and discuss the need to conduct studies with physiological relevance. For example, chloroquine was used in vivo (DOI:10.2174/187152609787847712) in humans in a double-blind controlled study, which eventually showed that chloroquine was not suitable for patients with CHIKV. In my view, this is the kind of information that the authors ought to include, discuss and posit upon in this review.

This manuscript reads, in parts, as one study per paragraph (with a compound per paragraph), where the summarise the data and conclude with “thus, compound x has antiviral activity by acting on y or z pathway ”.

Overall the manuscript is well-intentioned and the subject matter is very worthy of attention and a dedicated review topic. Minor English, grammar and punctuation/style errors need to be addressed, but overall the premise may fall short of what is needed for virologists, pathologists and immunologists and clinicians who work in the field of arboviruses, and especially chikungunya.  

Minor formatting issues needing attention (throughout the text, as some errors occur in multiple places).

- Chikungunya must be in lower case ‘c’ (as is seen in abstract and keywords)

P1L40: 'have raised concerns'

P1L41: 'Togaviridae' family

P1L41 'positive-sense, single-stranded RNA'

P1 L45: these may well be putative receptors, though recent studies indicate mxra8 acts as a mammalian host receptor for arthritogenic alphaviruses (PMID 31484075; PMID 31080063; PMID 31080061; PMID 29769725).

P2L46: 'directly translated into non-structural proteins, nSP1-4'

Figure 1: keep nomenclature consistent, with nSP (top left says NPS); Proteolysis (says protelysis) ; capsid disassembly (says diassembly)

P2L57: Cutaneous manifestations such as rash and pruritus/erythema are known to most arthritogenic alphaviruses, but "Digestive symptoms" should be clarified and cited.

P2L64: while it is true that several studies show antiviral activities of natural compounds against arbovirus replication, these are all (or almost all in vitro studies), and almost none show this in vivo. If I may suggest the authors to address this gap in understanding between in vitro and in vivo systems, and cover this in their review, this would greatly improve the manuscript.

P2L70 : shown to block CHIKV entry

P2L71: through lentiviral expression of chikv glycoprotein;; these compounds interfere with;; due to their effect on;;

P3L74: Khan and coworkers,

P3L76: against CHIKV, not 'the chikv'

P3L81: inhibited entry BY approx.

P3L83: mainly, not manly;;  prior to infection, not prior the infection

P3L98: Vero cells, not vero

Several other similar errors throughout the manuscript that require proofing.

Author Response

To the Reviewer 1,

First of all, thank you very much for taking the time to carefully evaluate the work we have submitted. We found the comments relevant and enlightening, guiding us towards a more coherent and satisfying work. We made some changes as suggested by all the reviewers. We also would like to clarify some points raised in your comments. Please, see below.

#1 'This manuscript reads more like a summary of specific studies than a compendium of work that has been done to investigate antiviral properties of natural compound'.

We are pleased with the comments and believe that they are very relevant to collaborate with our work. However, the purpose of this review is to do summarize compounds that have activity against Chikungunya in vitro that could provide information for further analysis in the development of novel antivirals.

#2 'Natural compounds have long been hailed as promising drug candidates - and they may well be - however at no point do the authors acknowledge and discuss the need to conduct studies with physiological relevance'.

We thank you the referee for this comment. As mentioned before, the aim of this review was to summarize in vitro research data. However, we agreed about the relevance to discuss studies with physiological relevance. We added an extra section, titled as “Prospective” where we include information concerning in vivo study.

#3 'Minor English, grammar and punctuation/style errors need to be addressed, but overall the premise may fall short of what is needed for virologists, pathologists and immunologists and clinicians who work in the field of arboviruses, and especially chikungunya'. 

We were extremely grateful for the description of the errors pointed out here. Language was revised as suggested.

#4 'Minor formatting issues needing attention (throughout the text, as some errors occur in multiple places). '

We performed all the changes as suggested above. A major revision of the text was performed to improve its comprehension and language was also revised.

We hope that with these modifications the manuscript will be acceptable for publication in your Journal.

With our bests,

Professor Ana Carolina Gomes Jardim

Reviewer 2 Report

The worldwide outbreak of the chikungunya virus (CHIKV) spreading by mosquito of Aedes sp. in the last years demonstrated the need of studies antivirals against CHIKV. Authors described that the virus was most recently, between 2007-2014, the virus was responsible for some cases in Europe and outbreaks in the America. The infection is associated to low rates of death however, it can progress to a chronic disease characterized by severe arthralgias in infected patients and is also associated with Guillain-barré syndrome. There is no specific antiviral against CHIKV. Treatment of infected patients is palliative and based on analgesics and non-steroidal anti-inflammatory drugs to reduce arthralgias. This review aims to summarize the natural compounds that have been revealed as potential antivirals against the chikungunya virus. There is no also specific antiviral or vaccine against CHIKV and treatment of infected patients is palliative and based on analgesics and nonsteroidal anti-inflammatory drugs to reduce arthralgias. In front of the lack of efficient anti- CHIKV therapy, research has been developed to identify new candidates to future treatments and antiviral based on natural molecules as a potential approach. That's why is very important to investigate the activity of antiviral drugs such as: Epigallocatechin gallate (Green tea), Chloroquine, Apigenin, Chrysin, Narigerin, Silybin and Prothipendyl, Flavaglines, Compounds from Tectona grandis,Trigocherrierin A, Harringtonine, Diterpene Ester (phorbol-12, 13-didecanoate, Aplysiatoxin related compounds, Daphanane diterpenoid ortho esters, Tannic Acid, Silymarin, Baicalein, Fisetin and Quercetagetin, Bryostatin, Prostatin, Berberine, Avermectin derivates.

            This review is a compendium and summary report for all this antiviral agents. I am definitely for the publication of this manuscript in Viruses.

Author Response

To the Reviewer 2,

First of all, thank you very much for taking the time to carefully evaluate the work we have submitted. We found the comments quite relevant and enlightening, guiding us towards a more coherent and satisfying work. We made some changes as suggested by all the reviewers. We hope that with these modifications the manuscript will be acceptable for publication in your Journal.

With our bests,

Professor Ana Carolina Gomes Jardim

Reviewer 3 Report

This  review thoroughly lists the natural antivirals currently being tested against CHIKV.  The authors do a very nice job summarizing the published studies. What is missing is a critical review of the published studies, and a perspective summarizing the field of natural antiviral. Also, the writing style in the review has a large number of choppy or short disconnected sentences which makes it difficult to read. 

Author Response

To the Reviewer 3,

First of all, thank you very much for taking the time to carefully evaluate the work we have submitted. We found the comments quite relevant and enlightening, guiding us towards a more coherent and satisfying work. We made some changes as suggested by all the reviewers. We also would like to clarify some points raised in your comments. Please, see below.

#1  ‘What is missing is a critical review of the published studies, and a perspective summarizing the field of natural antiviral. Also, the writing style in the review has a large number of choppy or short disconnected sentences which makes it difficult to read’. 

We made major revision of the text to improve its comprehension and language was also revised. Additionally, we added a topic with future perspectives for antiviral research.

We hope that with these modifications the manuscript will be acceptable for publication in your Journal.

With our bests,

Professor Ana Carolina Gomes Jardim

Reviewer 4 Report

   Recently, the outbreaks of Chikungunya virus have been one of public health problems all over the world. Therefore, the therapeutic antivirals have been needed against the virus infection. The authors reviewed the natural antivirals against Chikungunya, which examined in Laboratory. This review must be important and helpful for researchers in this area and also clinician for infectious diseases. I think, this paper should be published in the Journal. However, before publication, the authors need to correct and change little bit within the text as indicated below.

Line 50; Is “E3” correct? Line 87 and many lines; “(CHIKV-Rluc)” should be added after “Rluc”. Abbreviations must be shown when the word first appears at many parts in the text. Line 89 and many lines; The abbreviations like “IC50”, “MTT assay”, “SI” so on, must be explained just before use the words. Lines 92, 98, 115; “Vero” should be use instead of “vero”. Lines 185, 196, 197, 209, 214, 217, 227, 229, and 238; “EC50” should be use instead of “EC50”.  

Author Response

To the Reviewer 4,

First of all, thank you very much for taking the time to carefully evaluate the work we have submitted. We found the comments quite relevant and enlightening, guiding us towards a more coherent and satisfying work. We made some changes as suggested by all the reviewers. We also would like to clarify some points raised in your comments. Please, see below.

#1  ‘Line 50; Is “E3” correct? Line 87 and many lines; “(CHIKV-Rluc)” should be added after “Rluc”. Abbreviations must be shown when the word first appears at many parts in the text. Line 89 and many lines; The abbreviations like “IC50”, “MTT assay”, “SI” so on, must be explained just before use the words. Lines 92, 98, 115; “Vero” should be use instead of “vero”. Lines 185, 196, 197, 209, 214, 217, 227, 229, and 238; “EC50” should be use instead of “EC50”.

We performed all the changes as suggested above. A major revision of the text was performed to improve its comprehension and language was also revised. Additionally, we added a topic with future perspectives for antiviral research.

We hope that with these modifications the manuscript will be acceptable for publication in your Journal.

With our bests,

Professor Ana Carolina Gomes Jardim

Round 2

Reviewer 1 Report

Please find below recommendations on the revised version of the Manuscript( text in blue). I am happy with the revisions the authors have made, only a few (and very easy to address) corrections needed to ensure accuracy and legibility.

Q#2 'Natural compounds have long been hailed as promising drug candidates - and they may well be - however at no point do the authors acknowledge and discuss the need to conduct studies with physiological relevance'.

R#2 We thank you the referee for this comment. As mentioned before, the aim of this review was to summarize in vitro research data. However, we agreed about the relevance to discuss studies with physiological relevance. We added an extra section, titled as “Prospective” where we include information concerning in vivo study.

I acknowledge efforts (L408) to discuss some of the work that was done in vivo re: testing of antural compound-derived antivirals. Still, the abstract should be amended: last sentence should say “This review aims to summarize […] chikungunya virus in vitro"

Q#3 'Minor English, grammar and punctuation/style errors need to be addressed, but overall the premise may fall short of what is needed for virologists, pathologists and immunologists and clinicians who work in the field of arboviruses, and especially chikungunya'. 

R#3 We were extremely grateful for the description of the errors pointed out here. Language was revised as suggested.

Some minor English errors have appeared after correction:

Abstract: “natural compounds that have been described as antivirals against viruses SUCH AS Dengue, Yellow Fever etc..”

Abstract: “This review aims to summarize […] (remove ‘the’) chikungunya virus

L366: The correct genus name is “Berberis”, Family “Berberidaceae

L342: Please change the sentence to “Additionally, salicylate-derived analog 1, but not the other compounds, blocked entry of CHIKV pseudoparticles into BGM etc.”

L315: Please change the sentence to  “All three compounds were found to inhibit CHIKV replication etc..”

Author Response

To the Reviewer 1,

First of all, thank you very much again for taking the time to carefully re-evaluate the work we submitted. We made the changes as suggested by the reviewer. We also would like to clarify some points raised in your comments. Please, see below.

Q#1 I acknowledge efforts (L408) to discuss some of the work that was done in vivo re: testing of antural compound-derived antivirals. Still, the abstract should be amended: last sentence should say “This review aims to summarize […] chikungunya virus in vitro"

Answer: As suggested, we changed the last sentence adding “in vitro”.

Q#2 Some minor English errors have appeared after correction:

Abstract: “natural compounds that have been described as antivirals against viruses SUCH AS Dengue, Yellow Fever etc..”

Abstract: “This review aims to summarize […] (remove ‘the’) chikungunya virus

L366: The correct genus name is “Berberis”, Family “Berberidaceae

L342: Please change the sentence to “Additionally, salicylate-derived analog 1, but not the other compounds, blocked entry of CHIKV pseudoparticles into BGM etc.”

L315: Please change the sentence to “All three compounds were found to inhibit CHIKV replication etc..”

Answer: We are extremely grateful for the description of the errors pointed out here. Language was revised as suggested.